Capturing patient-reported area of knee pain: a concurrent validity study using digital technology in patients with patellofemoral pain

http://orcid.org/0000-0002-3054-3466 Matthews Mark 1 2
Rathleff Michael S. 3 4 5
Vicenzino Bill 1
http://orcid.org/0000-0002-8310-2744 Boudreau Shellie A. 5 sboudreau@hst.aau.dk
1 School of Health and Rehabilitation Sciences, Sports Injuries Rehabilitation and Prevention for Health Research Unit, University of Queensland , Brisbane, QLD , Australia
2 Sports and Exercise Science Research Institute, School of Sport, Faculty of Life and Health Sciences, University of Ulster , Belfast , UK
3 Research Unit for General Practice in Aalborg , Aalborg , Denmark
4 Department of Occupational Therapy and Physiotherapy, Department of Clinical Medicine, Aalborg University , Aalborg , Denmark
5 Center for Neuroplasticity and Pain, Centre for Sensory Motor Interaction, Department of Health Science and Technology, Faculty of Medicine, Aalborg University , Aalborg , Denmark
Hochheiser Harry
Electronic publication date: 2018 Mar 8
Publication date: 2018
Volume: 6
Electronic Location ID: e4406
Received 2017 Oct 24; Accepted 2018 Feb 2
Copyright: © 2018 Matthews et al.
Copyright year: 2018
Copyright holder: Matthews et al.
License: This is an open access article distributed under the terms of the Creative Commons Attribution License, which permits unrestricted use, distribution, reproduction and adaptation in any medium and for any purpose provided that it is properly attributed. For attribution, the original author(s), title, publication source (PeerJ) and either DOI or URL of the article must be cited.
License URL: https://creativecommons.org/licenses/by/4.0/

Keywords: Pain drawing, Tablet, Personal computer

Funding: University of Queensland Graduate School International Travel Award (GSITA), Australian Postgraduate Award Scholarship and the NHMRC Program 631717 Spar Nord Fonden Center for Neuroplasticity and Pain National Research Foundation DNRF121 Mark Matthews was supported by a University of Queensland Graduate School International Travel Award (GSITA), Australian Postgraduate Award Scholarship and the NHMRC Program Grant (Ref no: 631717). Spar Nord Fonden, Denmark supported the body schema template development. Shellie A. Boudreau is a part of Center for Neuroplasticity and Pain (CNAP) which is supported by the Danish National Research Foundation (DNRF121). The funders had no role in study design, data collection and analysis, decision to publish, or preparation of the manuscript.

==============================
Background

Patellofemoral pain (PFP) is often reported as a diffuse pain at the front of the knee during knee-loading activities. A patient’s description of pain location and distribution is commonly drawn on paper by clinicians, which is difficult to quantify, report and compare within and between patients. One way of overcoming these potential limitations is to have the patient draw their pain regions using digital platforms, such as personal computer tablets.

Objective

To assess the validity of using computer tablets to acquire a patient’s knee pain drawings as compared to paper-based records in patients with PFP.

Methods

Patients (N = 35) completed knee pain drawings on identical images (size and colour) of the knee as displayed on paper and a computer tablet. Pain area expressed as pixel density, was calculated as a percentage of the total drawable area for paper and digital records. Bland–Altman plots, intraclass correlation coefficient (ICC), Pearson’s correlation coefficients and one-sample tests were used in data analysis.

Results

No significant difference in pain area was found between the paper and digital records of mapping pain area (p = 0.98), with the mean difference = 0.002% (95% CI [−0.159–0.157%]). A very high agreement in pain area between paper and digital pain drawings (ICC = 0.966 (95% CI [0.93–0.98], F = 28.834, df = 31, p < 0.001). A strong linear correlation (R2 = 0.870) was found for pain area and the limits of agreement show less than ±1% difference between paper and digital drawings.

Conclusion

Pain drawings as acquired using paper and computer tablet are equivalent in terms of total area of reported knee pain. The advantages of digital recording platforms, such as quantification and reporting of pain area, could be realized in both research and clinical settings.

Introduction

Pain drawings that capture area, location and distribution of pain can be used to aid diagnosis and track changes over time (e.g. after a course of treatment) (Margolis, Tait & Krause, 1986; Abbott et al., 2015; Southerst et al., 2013; MacDowall et al., 2017). Pain drawings are commonly captured on paper-based body schemas, charts or sketched diagrams (Thompson et al., 2009; Wood et al., 2007; Sengupta et al., 2006; Creamer, Lethbridge-Cejku & Hochberg, 1998; Post & Fulkerson, 1994; Elson et al., 2011). Despite studies reporting paper-based pain drawings to have good to very good inter-rater and intra-rater reliability, they present limitations for clinicians to easily quantify and compare pain areas within and between patients (Thompson et al., 2009; Wood et al., 2007; Sengupta et al., 2006; Creamer, Lethbridge-Cejku & Hochberg, 1998; Post & Fulkerson, 1994; Elson et al., 2011). One way of overcoming these limitations is to have the patient draw the area and location of their pain on digital platforms, such as personal computer (PC) tablets. PC tablets offer considerable advantages to patients in health care settings. The advancements of touch-screen technology, such as those employed in smart-phones and hand-held PC tablets make it possible and easier to acquire, quantify, report and compare patient-completed pain drawings.

Recent studies have investigated pain drawings recorded on digital platforms and shown reliability commensurate with that of paper drawings (Boudreau et al., 2016; Boudreau, Kamavuako & Rathleff, 2017). A study using high-resolution and contoured (3D) images of the knee found a high proportion of patients with patellofemoral pain (PFP) reported knee pain in mirrored locations and that the pain drawings were exceptionally symmetrical (Boudreau, Kamavuako & Rathleff, 2017). Interestingly, when the knee pain drawings were compared to duration of symptoms, those with a longer duration of symptoms appeared to draw patterns more of an ‘O’ shape. Given that longer duration of PFP symptoms has been shown to be a prognostic indicator of a poor outcome (Matthews et al., 2016), capturing pain areas digitally will allow quantification and real time insight into the patient’s condition, which may optimise patient management.

The aim of this study was to assess the concurrent validity of paper and digital pain drawings in patients with PFP. The hypothesis was that the area of knee pain in patients with PFP acquired using hand-held PC tablets (digital drawings) is equivalent to using pen and paper.

Methods

This concurrent validity study investigated the agreement of pain area between patient completed paper and digital pain drawings. To minimise order and learning effect bias, the order of completing paper and digital pain drawings was randomised with approximately 1–2 min between drawings. The reporting of the study follows the Guidelines for Reporting Reliability and Agreement Studies (Kottner et al., 2011).

Participants

A consecutive sample of 35 patients from a clinical trial (Matthews et al., 2017) were recruited from the community of Aalborg, Denmark via public advertising or referred from sports medicine clinics and general practitioners. A musculoskeletal physiotherapist with experience in managing patients with PFP screened participants for inclusion into the study. Inclusion criteria were: (1) aged between 18 and 40 years with a history of non-traumatic anterior retro or peripatellar knee pain that was greater than six weeks duration, (2) self-reported worst pain over the previous week equal to or greater than 3 out of 10 on a numerical pain scale (0 = no pain, 10 = worst pain imaginable), (3) symptoms provoked by at least two of the following activities: squatting, running, stair ascending or descending or prolonged sitting were included. Individuals were excluded if they had any one of the following: concomitant injury or pathology of other knee structures (e.g. ligament, meniscal, tendon, iliotibial band, pes anserinus, fat pad), or a history of knee surgery, patellofemoral dislocation or subluxation, Osgood–Schlatter’s disease, Sinding–Larsen–Johansson syndrome, a positive patellar apprehension test or evidence of knee joint effusion. The Ethics Committee in the North Denmark Region and the Danish Data Agency approved the study (N-20140022). All participants were provided with verbal and written information about the procedures of the study, and written informed consent was obtained prior to data collection.

Data collection

The participants were instructed by a second musculoskeletal physiotherapist to complete in a randomised order a paper and digital pain drawing to the best of their abilities. The verbal instruction given to the patients for both the paper and digital pain drawings was ‘please use the pen to draw on the paper/screen where you most often experience your knee pain’. Digital pain drawings were performed on a PC tablet (Samsung Galaxy Note 10.1, Android 4.1.2) that displayed a 3D body schema of the lower torso (from the anterior superior iliac spine prominences, and below, such that contours of the left and right legs and knees were clearly visible) (Fig. 1). Participants used a permanent red marker with a 1 mm thick felt tip (Edding 400, Wunstorf, Germany) for the paper drawings and an S Pen™ that accompanied the tablet so as to control for line thickness and to enable precise drawings. In order to enable a valid comparison (i.e. like with like) three a priori calibrations were performed: (i) the thickness of the line created by the S Pen™ on the tablet was set to equal the thickness of the permanent red marker on the paper, (ii) the S-Pen drawing on the digital device created red ‘pixels’ to indicate the patient’s knee pain location, area and distribution and (iii) the image of the lower body schema as displayed on paper was scaled to the same size as the lower body schema displayed on the computer tablet.

Figure 1 Unmarked 3D lower body leg schema on paper (A) and digital (B).

Sample-size

The hypothesis is that a hand-held PC tablet is a comparable and equally valid method for acquiring patient-completed pain drawings as compared to pen and paper records. Based on a previous study (Boudreau et al., 2016), it is hypothesized that the difference in pain area between these methods will not be greater than 1%. We used an equivalency sample-size calculation to determine the sample-size required to show that there was no clinically relevant differences between pain area(s) collected by paper vs. pain area(s) collected using a computer tablet. The data used for the sample-size calculation were based on means and standard deviations of two PFP groups collected in a pilot study. Using a conservative correlation factor of 0.5 between drawings, an equivalency limit of 6,720 pixels and a SD of 4,451 pixels at 5% significance and 95% power, it was necessary to collect 35 paper-digital pairs of pain drawings.

Data management and analysis

To compare pain areas between paper and digital records, we calculated the pixel density of the scanned paper and digitally acquired pain drawings. The pixel densities of the area of the pain drawing on both paper and digital media were expressed as a percentage of the total area of a blank body map of the lower body schema (i.e. a reference standard). Any pain areas that were ambiguous in terms of the boundaries and extent of pain were excluded, such as cross-hatching with unfilled areas, as shown in Figs. 2A–2C.

Figure 2 The excluded pain drawings, which did not follow the drawing instructions such as the use of arrows (A), circles and scribbled lines (B) and zigzag lines (C).

Assessment of pain area for paper drawings

One investigator (MM) who was not involved in data collection and was blind to the computer tablet records processed the paper records to determine pixel density of the paper-based pain recordings. This investigator scanned all the paper records for subsequent determination of pixel density from the digital record.

Paper drawings were scanned at 300 ppi, saved as a PDF file and imported into Adobe Photoshop CC (2015.1; Adobe Systems, San Jose, CA, USA) for analysis. The pen selection function was used to trace a path of the body schema and pain area to create a ‘selected area’ from which the pixel density was calculated. First, a reference standard of pixel density for the paper version of the lower body schema was created by scanning an unused paper version of the lower body schema. The total pixel density was calculated three times and then averaged. The pain area for each participant’s paper record was traced and pixel density calculated.

Assessment of pain area of digital drawings

The Navigate Pain™ software that was preloaded on the PC tablet and automatically calculates the red pixels associated with the pain drawings. The red pixels are also expressed relative to the total pixel area (total drawable area) of the lower body schema. The percent and absolute number of pixels were exported directly into an excel document for data analyses.

Data analysis

One-sample t-tests were used to compare the difference in pixel density between paper and computer table recordings of the patient’s pain drawings. Intraclass correlation coefficient (ICC) using absolute-agreement, two-way mixed model was used to determine the agreement between paper and digital platforms. Pearson’s correlation coefficients were used to express the degree of linear association between the two methods (Koo & Li, 2017). Limits of agreements (LoAs), using Bland–Altman plots, were used to express the agreement between the paper and computer tablet methods. The LoAs were presented as a range indicating the maximal potential difference between the two methods in 95% of the ratings. All statistics were performed in IBM SPSS Statistics, version 24 (IBM Corp., Armonk, NY, USA) and α = 0.05 was used as level of significance.

Results

Thirty-five participants were recruited into the study. Three participants were excluded, as they did not follow the drawing instructions (Fig. 2). One participant was excluded due to the use of arrows in their paper drawing to indicate a pain area (Fig. 2A). One participant was excluded due to their paper drawing having incomplete circles with scribbled lines, leaving it unclear if it truly represents their pain area (Fig. 2B). One participant was excluded due to the ambiguous use of zigzag lines in both the paper and digital to indicate pain area (Fig. 2C). The remaining 32 participant drawings were analysed (Fig. 3). Participants were predominantly female (78%), mean age of 24.5 (5.6) years old, BMI of 23.7 (3.4) and with average symptom duration of 69.7 (2–192) months. Twenty-five of the 32 participants reported and marked bilateral symptoms.

Figure 3 The variability of the 32 digital knee pain drawings, from patients diagnosed with PFP, used to assess pain area between paper and digitally acquired drawings.

A very high agreement in pain area between paper and digital pain drawings as reflected by an ICC of 0.966 (95% CI [0.93–0.98], F = 28.834, df = 31, p < 0.001). There was a strong linear correlation in pain area between paper and digital pain drawings (R = 0.93, p < 0.0001) (Fig. 4). The drawings with the largest difference (1.1%) and smallest difference (0.05%) in pain area between paper and digital pain drawings are depicted in Fig. 5.

Figure 4 A strong linear correlation in pain area between paper and digital pain drawings.

Figure 5 Pain drawings associated with the smallest (0.05%, A) and largest (1.1%, B) differences in pixel density between paper and digitally acquired pain drawings.

No significant difference was found between the two methods (p = 0.98), with the mean difference and 95% CI (mean difference = 0.002% (95% CI [−0.159–0.157%]) and the Bland–Altman plot (Fig. 6) all found to be lower than the pre-defined ±1% equivalence margins.

Figure 6 Bland–Altman plot showing the limit of agreement in pain area between paper and digital pain drawings.

Discussion

This study found minimal differences in pain area recordings made on paper or a PC tablet. Results indicate that any difference in area would likely be less than ±1%. These results support the hypothesis and provide an important first step towards validation of digitally acquired pain drawings for pain assessment in the knee.

Digital pain drawings offer the advantages of being easily acquired, quickly quantified and interpreted to assist in clinical diagnosis and comparison over time. Previous studies have investigated paper-based pain mapping to assist in clinical diagnosis of knee and shoulder pain (Elson et al., 2011; Bayam et al., 2011, 2017). Participants were instructed to use small crosses (‘X’) (Elson et al., 2011) or symbols (Bayam et al., 2011, 2017) to mark out their pain location, type, distribution and severity and several ‘X’s if pain was present in more than one location. Once the drawing was made, a grid-like template was use for categorizing anatomical zones of the knee. In its simplest form, placement of ‘X’s allow quick reporting and the identification of a general location of the pain. However, the utilisation of ‘X’s to mark out pain area limits the accuracy of the patient expressing their pain distribution and raises doubt on the diagnostic utility. By using a digital method in this current study, patients were able to fully express their perceived pain location and distribution and not be restricted to simple ‘X’s. Although not a focus of the present study, digital drawings could improve the diagnostic accuracy of the pain drawings and be potentially useful and cost-effective as an adjunct tool to quantify and interpret a patient’s pain.

An unexpected observation in this study was the variability of the individual pain drawings in a cohort with a homogenous diagnosis. Of the 32 participants, 23 (72%) drew pain areas on both knees with 12 (38%) patients drawing pain in two or more locations in the same knee. This observation has also been seen in previous studies (Thompson et al., 2009; Elson et al., 2011). Thompson et al. (2009) reported patients indicating pain in two or three locations or two regions with several participants drawing three areas of local pain. By drawing multiple areas, it could appear that participants are expressing diffuse symptoms, multi-location or different pain types, e.g. sharp or aching types of pain. The variability of these drawings could also be a reflection of the heterogeneous nature of PFP. PFP is an often persistent, multifactorial condition that is diagnosed by its clinical presentation with exclusion of other conditions (Crossley et al., 2016). The simplified approach of this diagnosis could lead to the captured pain drawings expressing a variety of local nociceptive, peripheral and central sensitization pain presentations. Whilst this current study compared the percentage of pain area, the change in location of this area between drawings was not assessed. A change in location between two pain drawings has yet to be assessed, even when considering the earliest reliability studies. A change in location would of course be bodily context dependent. With more or less acceptable deviations in location change dependent on the pain being assessed such as the knee or low back. The variability of the present drawings also warrants further consideration in future studies, looking at the relationship between patient-perceived pain drawings, location and diagnosis.

The level of anatomical detail displayed on the body schema used in the current study may have contributed to the minimal difference obtained between paper and digital pain drawings. In a pain-mapping study on a chronic neck pain cohort, results found high reproducibility between paper and digital platforms as well as between simple body outlines and high-resolution contoured body schemas (Boudreau et al., 2016). However, a small fixed negative bias was identified with slightly smaller drawings performed on paper than PC tablet, and pain areas were drawn slightly larger on the less-detailed body outline in comparison to the high-resolution body schemas (Boudreau et al., 2016). One explanation for these findings could be the greater level of anatomic detail of the body schema being more recognizable to the patient. When a patient is able to see important anatomical landmarks, greater accuracy and precision of the pain drawings may occur.

A key consideration of this study is the verbal instructions given to the participants in the study. The instructions given may have allowed some degree of ambiguity. This is evident by three of the 35 excluded pain drawings. The pain drawings were excluded due to the amounts of unmarked areas within circles and use of zigzag lines for shading in larger areas. As a recommendation for future studies the instruction set should be of explicit clarity and possibly, include a sample example of a correct and incorrect pain drawing. For example, ‘please draw on the image that best represents the location and area of your pain. Please use solid lines or completely filled in areas, leaving no clear spaces within the area’.

A second consideration for this study was the method of acquiring the pain drawings. Clinicians have traditionally completed pain drawings on body schemas, charts or sketched diagrams (Thompson et al., 2009; Wood et al., 2007; Sengupta et al., 2006; Creamer, Lethbridge-Cejku & Hochberg, 1998; Post & Fulkerson, 1994; Elson et al., 2011). However, considerations were identified in these studies that warranted attention. A comparison study found that clinicians drew significantly smaller areas of pain when compared to the patients’ drawing (Post & Fulkerson, 1994), suggesting observer bias and filtering of information by the clinician which may not accurately represent the patient’s report. In the current study, this consideration was addressed by asking patients to complete the drawings. It is imperative future studies of pain drawings, and indeed clinical utilization, ensure patient-completed pain drawings in guiding the diagnosis (Post & Fulkerson, 1994).

The result of this study opens possibilities on the benefits of using digital platforms in clinical examination. By combining touch-screen technology with a high-detailed body schema, pain drawings can be quickly quantified and interpreted to facilitate clinical decisions. In turn, clinicians can easily monitor symptoms by comparing pain drawings within and between patients over time. With continued development of software, new avenues could be created for research. Future studies could explore and interpret pain drawings to enable identification of previously unknown pain patterns. Identification of pain patterns could be particular pertinent in patients with persistent and prevalent conditions such as knee or low back pain. Several studies on patients with low back pain have used pain drawings to locate body regions where patients have experienced pain (Hullemann et al., 2017; Gerhardt et al., 2016). Results suggest pain drawings might help to understand the patient’s underlying mechanism of pain and improve treatment outcomes (Hullemann et al., 2017; Gerhardt et al., 2016). The advantages offered by digital recording platforms, such as automatic quantification and reporting of pain area, could be realized in both clinical settings and research to improve healthcare.

Conclusion

This study found knee pain drawings acquired on digital and paper-based platforms to be comparable in area. This study provides a first important step for testing of a digital interface that can facilitate the precise communication of patient-perceived knee pain area and location. The use of digital technology in health care, including digital platforms of patient-perceived pain drawing records, opens up many exciting possibilities in clinical and research settings.

Supplemental Information

Supplemental Information 1 Raw data.

Click here for additional data file.

We thank Richard Spence for bulk extraction of the pixel density calculations from the Navigate Pain software and for technical support. We thank Mette Bøgedal for screening and including the participants and Lukasz Winiarski for data collection.

Additional Information and Declarations

Competing Interests

Author Contributions

Human Ethics

Data Availability

Shellie A. Boudreau is a co-developer of the Navigate pain™ software application used in this study. All other authors declare that they have no competing interests.

Mark Matthews conceived and designed the experiments, performed the experiments, analysed the data, contributed reagents/materials/analysis tools, prepared figures and/or tables, authored or reviewed drafts of the paper.

Michael S. Rathleff conceived and designed the experiments, analysed the data, contributed reagents/materials/analysis tools, prepared figures and/or tables, authored or reviewed drafts of the paper.

Bill Vicenzino conceived and designed the experiments, analysed the data, contributed reagents/materials/analysis tools, prepared figures and/or tables, authored or reviewed drafts of the paper.

Shellie A. Boudreau conceived and designed the experiments, performed the experiments, analysed the data, contributed reagents/materials/analysis tools, prepared figures and/or tables, authored or reviewed drafts of the paper.

The following information was supplied relating to ethical approvals (i.e. approving body and any reference numbers):

The Ethics Committee in the North Denmark Region and the Danish Data Agency approved the study. N-20140022.

The following information was supplied regarding data availability:

The raw data has been supplied as Supplemental Files.

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
