# Peer review of "Capturing patient-reported area of knee pain: a concurrent validity study using digital technology in patients with patellofemoral pain"

_PeerJ, doi:10.7717/peerj.4406_

## Round 0.1 · original submission · Minor Revisions

The reviewers have made several comments which should be relatively easy to address with minor revisions and some limited additional analyses. Please respond to these comments in your revision.

·

Basic reporting

L45: The GRRAS guidelines recommend a description of what is already known about reliability in the introduction. Since this study was not investigating reliability, it would seems appropriate to demonstrate that the paper and digital methods are reliable.

Experimental design

No Comment

Validity of the findings

No Comment

Additional comments

The aim of this study was to determine the validity of digital drawings of pain area compared to paper drawings. This study is very simple and well-designed, and the manuscript is well written. This topic is important since digital drawings might be a more efficient method of collecting this information both clinically and in research.

Specific Comments

L91: Did the order the participants completed the drawings influence the percent difference between the drawings? It would seem that one might become more confident in what they draw the second time around (like a learning effect). Looking at the raw data, it is almost evenly split between which had a greater percent (just subtracting the two values) paper or digital. Could the order have influenced the findings. This likely would be a limitation or something discussed later.

L151-152: Were participant drawing only excluded from the digital drawings? This is what it looks like from the figures but were their paper drawings similar? Could this be a limitation?

L167: Please provide the actual percent difference for the largest and smallest.

L177: This study only compared percent areas; however, this does not assess the location of the pain. I would be appropriate to include a limitation that this study did not investigate validity of the location, which would be clinically important if different.

·

Basic reporting

The article is nicely written. In the following areas, minor changes to the English language should be made, suggestions included:

Line 108-9; replace tablets with tablet
Line 120: replace density with densities
Line 152: please replace “their ambiguity of” with “the ambiguity of”
Line 186: please replace “X” with “X”s
Line 186: please replace it’s with its
Line 193: please replace “adjunct tool” with “as an adjunct tool”
Line 201: please replace i.e. with e.g.
Line 223, remove comma

Lines 182 to 194. You may want to describe the placement of “X”s as identifying the pain area by marking squares on a grid.

Experimental design

An interesting study, with a well considered design.

Lines 144-5: Please report the intraclass correlation coefficient (ICC) as opposed to Pearson / R-squared as this tests not just for a linear trend but that values are equal (i.e. gradient = 1).

Line 148, please provide the reference for SPSS.

Validity of the findings

Line 155: please provide mean and SD for BMI.

Lines162-4. As this is an equivalence study, then the test of means is not on it's own sufficient - as identifying no significant difference does not enable us to conclude definitively that the means are the same. The correct analysis should be conducted by comparing the 95% CI for the mean difference and seeing whether this lies within or outside the acceptance region – which I understand to be +/- 1% from your sample size calculation (line 111). You quote the 95% CI as -0.159 to 0.157. The units are unclear, but if this is -0.159% to 0.157% then your confidence interval for the mean difference falls entirely within the acceptance region of -1.0% to +1.0% and so it is possible to conclude equivalence based on your pre-defined acceptance criterion. Please clarify this point, and adjust the text describing the results appropriately. More details of this approach can be found in Byrom and Tiplady. ePRO: electronic solutions for patient-reported data. Gower. pp189-90.

Lines 165-8. I agree that a linear correlation is present. Please be aware that Pearson’s correlation tests for a linear trend, but in an equivalence study we are looking to demonstrate that values are equal – i.e. to see a linear trend with a gradient of 1.0. This is more appropriately assessed by using the intraclass correlation coefficient (ICC). Please report the ICC instead of R-squared.

Lines 236 – 248. In conclusion, it would seem a good future extension to automate the calculation of the pain area (pixel density) using an algorithm to interpret the drawing on the tablet if the approach is taken forward.

Additional comments

This is an interesting study with good potential for application in routine care and clinical trials.

---

## Round 0.2 · accepted · Accept

Thanks for diligently addressing the comments of the reviewers.

·

Basic reporting

The author's revisions have satisfied this reviewer's requests.

Experimental design

The author's revisions have satisfied this reviewer's requests.

Validity of the findings

The author's revisions have satisfied this reviewer's requests.

Additional comments

The author's revisions have satisfied this reviewer's requests.

·

Basic reporting

No comment.

Experimental design

No comment.

Validity of the findings

No comment.

Additional comments

Thank you for making the suggested revisions to your paper. It is a well written and interesting piece.